# Tumor-Infiltrating CD20^+^ B Lymphocytes: Significance and Prognostic Implications in Oral Cancer Microenvironment

**DOI:** 10.3390/cancers13030395

**Published:** 2021-01-21

**Authors:** Faustino Julián Suárez-Sánchez, Paloma Lequerica-Fernández, Juan Pablo Rodrigo, Francisco Hermida-Prado, Julián Suárez-Canto, Tania Rodríguez-Santamarta, Francisco Domínguez-Iglesias, Juana M. García-Pedrero, Juan Carlos de Vicente

**Affiliations:** 1Department of Pathology, Hospital Universitario de Cabueñes, 33394 Gijón, Asturias, Spain; faustinosuarezsanchez@gmail.com (F.J.S.-S.); juliansuarezcanto@gmail.com (J.S.-C.); dominguezifrancisco@uniovi.es (F.D.-I.); 2Department of Biochemistry, Hospital Universitario Central de Asturias (HUCA), C/Carretera de Rubín s/n, 33011 Oviedo, Asturias, Spain; paloma.lequerica@sespa.es; 3Instituto de Investigación Sanitaria del Principado de Asturias (ISPA), Instituto Universitario de Oncología del Principado de Asturias (IUOPA), Universidad de Oviedo, C/Carretera de Rubín s/n, 33011 Oviedo, Asturias, Spain; jprodrigo@uniovi.es (J.P.R.); UO177476@uniovi.es (F.H.-P.); tania.rodriguez@sespa.es (T.R.-S.); 4Department of Otolaryngology, Hospital Universitario Central de Asturias (HUCA), C/Carretera de Rubín s/n, 33011 Oviedo, Asturias, Spain; 5Department of Surgery, University of Oviedo, 33006 Oviedo, Asturias, Spain; 6Ciber de Cáncer (CIBERONC), Instituto de Salud Carlos III, Av. Monforte de Lemos, 3-5, 28029 Madrid, Spain; 7Department of Oral and Maxillofacial Surgery, Hospital Universitario Central de Asturias (HUCA), C/Carretera de Rubín s/n, 33011 Oviedo, Asturias, Spain

**Keywords:** CD20, tumor-infiltrating B lymphocytes, prognosis, oral squamous cell carcinoma, immunohistochemistry

## Abstract

**Simple Summary:**

The complex interplay between the different cellular components in the tumor microenvironment (TME) dynamically modulates the antitumor immune response. This study investigates the prognostic relevance of CD20^+^ tumor-infiltrating B lymphocytes in oral squamous cell carcinoma (OSCC), and also possible relationships with other immune subtypes and key players within the oral TME.

**Abstract:**

Immunohistochemical analysis of stromal/tumoral CD20^+^ B lymphocytes was performed in 125 OSCC patients. Correlations with immune profiles CD4^+^, CD8^+^, and FOXP3^+^ tumor-infiltrating lymphocytes (TILs), tumoral PD-L1, and stem-related factors NANOG and SOX2 were assessed, and also associations with clinical data and patient survival. There was a strong positive correlation between the infiltration of CD20^+^ B lymphocytes and other immune profiles (i.e., CD4^+^, CD8^+^, and FOXP3^+^ TILs, and CD68^+^ and CD163^+^ macrophages) both in stroma and tumor nests. Strikingly, CD20^+^ TILs were inversely correlated with NANOG/SOX2 expression. Stromal CD20^+^ TILs were significantly associated with T classification and second primary tumors. A stratified survival analysis showed that tumoral CD20^+^ TILs were significantly associated with prognosis in male and younger patients, with tobacco or alcohol consumption, high tumoral CD8^+^ TILs, low tumoral infiltration by CD68^+^ macrophages, positive PD-L1 expression, and negative NANOG/SOX2. Multivariate Cox analysis further revealed clinical stage and tumoral CD20^+^ TILs independently associated with disease-specific survival (HR = 2.42, *p* = 0.003; and HR = 0.57, *p* = 0.04, respectively). In conclusion, high CD20^+^ TIL density emerges as an independent good prognostic factor in OSCC, suggesting a role in antitumor immunity. This study also uncovered an inverse correlation between CD20^+^ TILs and CSC marker expression.

## 1. Introduction

Oral squamous cell carcinoma (OSCC) is a heterogeneous malignant disease whose complex and dynamic progression could be the result of multiple genetic or epigenetic alterations among distinct cell types within the tumor as well as the surrounding tumor microenvironment (TME) [1,2]. In the TME, besides stromal cells, there are various populations of tumor-infiltrating lymphocytes (TILs), including B cells which may account for up to 25% to 40% of all cells in different tumor types [3]. These data suggest that B cells may play crucial roles in antitumor immunity in cooperation with other immune cells, such as T lymphocytes and macrophages [4]. Tumor-infiltrating lymphocytes in the TME may alter tumor biology, and the type, density, and location of these cells could influence tumor progression. The number of TILs in the TME may be explained by three possibilities. Firstly, the number of immune cells is associated with the immunogenicity of a specific tumor which may induce local activation and proliferation of immune cells. Secondly, an increase in B lymphocytes during tumorigenesis and tumor progression may be due to an increase in the load of associated antigens on tumor cells, antigens that can induce local activation and proliferation of the immunocytes. Thirdly, the increase in TIL number may reflect enhanced cytokine production by the tumor cells [5]. Cells of innate immunity, mainly tumor-associated macrophages (TAMs) can be polarized into antitumoral M1 macrophages, expressing CD68, and protumoral and immunosuppressive M2 macrophages, characterized by coexpression of CD68 and CD163 [6]. At the same time, the adaptive immune response is orchestrated by T and B lymphocytes. T cells are divided into CD8^+^ T cells and CD4^+^ T helper (Th) cells [7]. CD8^+^ cytotoxic T lymphocytes (CTLs), usually supported by CD4^+^ T helper 1 (Th1) cells [8], are considered the major effector immune cells directed against tumor cells [9]. On the contrary, regulatory T cells (Tregs), a subset of Th cells that express the transcription factor FOXP3, take part in the immune tolerance by suppressing self-antigen reactive T cells, thereby promoting the immune evasion of cancer [10].

Cancer cells express modified proteins or tumor-specific antigens that usually elicit an immune response [11]. Upon binding of an antigen, and supported by cytokines released by Th cells, B cells can differentiate into plasma cells and produce antibodies specific to the antigen that trigger their differentiation [7]. Beyond their role in humoral immunity, B cells can also modulate T cell responses to antigens [12]. In the T-helper 1 (Th1)/Th2 paradigm, the activity of cytotoxic T cells is supported by the Th1 lineage and M1 macrophages, while contrarily Tregs, B lymphocytes, and M2 macrophages are more closely related to the tumor-promoting Th2 response [13]. Programmed cell death ligand-1 (PD-L1) is a cell-surface glycoprotein that induces T-cell anergy and apoptosis by activating the PD-1 receptor on T lymphocytes [14,15]. Mizoguchi et al., [16] have described a subpopulation of B cells called regulatory B cells (Bregs) that exhibit tumor-promoting effects by their ability to suppress T cell responses through the secretion of cytokines such as IL-10 and TGF-β as well as to upregulate immune-regulatory ligands such as PD-L1, inducing CD4^+^ T cell death through the expression of FasL [17,18,19].

According to the cancer stem cells (CSCs) model, tumors are composed of heterogeneous cellular components including a rare subpopulation (0.01–10% of cells within the tumor [18]) exhibiting CSC self-renewal and plasticity. This subset of CSC pluripotent cells has been demonstrated to play crucial roles in tumor initiation, progression, aggressiveness, heterogeneity, metastasis, recurrence, and treatment resistance [19,20,21]. CSCs bear stemness properties supported by gene master regulators, such as NANOG and SOX2 [22], involved in OSCC tumorigenesis, poor differentiation, and bad prognosis [23,24,25]. Furthermore, CSCs possess immunoregulatory properties [26] and can inhibit CD8^+^ T cells or induce Tregs and myeloid-derived suppressor cells (MDSCs) [27].

From a prognostic point of view, stromal expression of CD163^+^ macrophages in head and neck squamous cell carcinomas (HNSCCs) [6], and infiltrating T lymphocytes in OSCC have been consistently relevant [28,29]. However, the role of infiltrating B lymphocytes in these carcinomas has not been fully clarified [30], as both positive and negative impacts of B cells on tumor progression and prognosis have been reported [31,32].

CD20, a pan B-cell marker, is a membrane-embedded phosphoprotein, encoded by the *MS4A1* gene, expressed in B lymphocytes, and typically lost when B cells become plasma cells. This marker has been extensively and widely used for the evaluation of inflammatory infiltrate in different solid tumors [33]. We herein investigated the clinical relevance of infiltrating B lymphocytes, by means of CD20 immunohistochemical evaluation in both the stroma and tumor nests using a large homogeneous cohort of 125 OSCC specimens. Correlations with clinicopathological features and impact on patient prognosis were assessed, and also possible relationships with other important cellular components in the oral TME, such as immune subtypes (i.e., T cells, macrophages), as well as tumoral PD-L1 expression and two important CSCs-related factors NANOG and SOX2.

## 2. Results

### 2.1. Immunohistochemical Analysis of CD20^+^ TILs in OSCC Tissue Specimens and Associations with Other Immune Subtypes

The mean number of CD20^+^ B cells in the tumor nests and the surrounding stroma was 18.67 ± 1.63 per mm^2^ (range: 0.00 to 18.67) and 42.47 ± 78.63 per mm^2^ (range: 0.00 to 426.67), respectively. Representative images of stromal and tumoral CD20^+^ TILs detected in OSCC specimens are shown in Figure 1. The mean numbers for other immune subtypes, such as CD4^+^ and CD8^+^ TILs, FOXP3^+^ Tregs, CD68^+^, and CD163^+^ macrophages are summarized in Appendix A.

There was a strong positive correlation between the infiltration of CD20^+^ B cells, cytotoxic CD4^+^ and CD8^+^ T cells, regulatory FOXP3^+^ T cells, and CD68^+^ and CD163^+^ macrophages in both stroma and tumor nests (Table 1). We also performed a hierarchical clustering analysis, shown in Figure 2A. These data further strengthen the close relationship between B cell and T cell infiltration. In particular, tumoral and stromal CD20^+^ B cell infiltration was more closely related to cytotoxic CD4^+^ and CD8^+^ TILs.

### 2.2. Associations between CD20^+^ TILs and Clinicopathological Variables

Stromal CD20^+^ TIL infiltration was significantly associated with T classification and with the occurrence of second primary tumors. In fact, the mean number of stromal CD20^+^ B cells was significantly lower in T3 and T4 tumors compared with T1 and T2, as well as in those patients who did not develop a second primary oral carcinoma (Table 2). Hierarchical clustering analysis is also shown in Figure 2B, further indicating that CD20^+^ B cell infiltration was closely related with the development of a second primary tumor, NANOG and SOX2 expression or the presence of tumor recurrence, whereas expectedly T, grade, N, and stage were closely related, or tobacco and alcohol consumption.

Stromal CD20^+^ B cell infiltration was not associated with any of the remaining clinicopathological parameters studied. Tumoral CD20^+^ infiltration was not associated with any clinicopathological variables. It was not possible to calculate the ratios of infiltrating CD20^+^ B cells and the remaining TILs and macrophage markers for all the cases as various markers showed negative expression (scored as 0). In fact, stromal/tumoral CD20^+^/CD8^+^ ratios could thus be respectively determined in 125 and 118 cases, stromal/tumoral CD20^+^/CD4^+^ ratios respectively determined in 125 and 95 cases, stromal/tumoral CD20^+^/FOXP3^+^ ratios in 106 and 77 cases, stromal/tumoral CD20^+^/CD68^+^ ratios in 125 and 123 cases, and finally, stromal/tumoral CD20^+^/CD163^+^ ratios were determined in 125 and 122 cases, respectively. Stromal CD20^+^/CD8^+^ ratio was significantly associated with T classification and second primary tumors, being consistently lower in T3 and T4 tumors and in patients who developed second primary malignancies. The stromal CD20^+^/CD4^+^ ratio was significantly associated with the development of second primary tumors. Stromal and tumoral CD20^+^/FOXP3^+^ ratios were significantly associated with patient age, with both being lower in patients younger than 65 years. In addition, the stromal CD20^+^/FOXP3^+^ ratio was significantly associated with tumor recurrence, being lower in non-recurrent cases (Table 3).

Stromal CD20^+^/CD68^+^ and CD20^+^/CD163^+^ ratios were significantly and consistently associated with T classification, tumor grade, and second primary tumors. Both ratios were lower in T3 and T4 tumors, and in cases that did not develop a second primary tumor. In marked contrast, while the stromal CD20^+^/CD68^+^ ratio was higher in well-differentiated tumors, the stromal CD20^+^/CD163^+^ ratio showed a higher value in moderate and poorly differentiated tumors (Appendix A).

### 2.3. Associations between CD20^+^ TILs, CSC Markers, and PD-L1

Positive staining of SOX2 and NANOG were respectively detected in 49 (40%) and 39 (32%) cases, as previously reported [24,25]. Noteworthy, we found concordant results when assessing the correlation between CD20^+^ B cell infiltration and the expression of these two CSC markers. The mean number of CD20^+^ TILs in both the tumor and the surrounding stroma was consistently higher in tumors harboring negative expression of SOX2 and NANOG, although this inverse relationship only reached statistical significance for the SOX2 marker (*p* = 0.008) (Table 4).

Tumoral PD-L1 expression in more than 10% of tumor cells was previously defined as clinically relevant [34] and was detected in 18 (15%) cases in our OSCC cohort. CD20^+^ infiltrating B cells in the tumor nests were higher in positive PD-L1 tumors, whereas stromal CD20^+^ TILs showed a higher density in negative PD-L1 tumors (i.e., expression in less than 10% of tumor cells). However, none of these relationships were statistically significant (Table 4).

### 2.4. Impact of CD20^+^ TIL Infiltration on the Survival of OSCC Patients

Follow-up data were available for 121 patients (range 6–230 months, mean 74 and median 61 months). Patients with smaller (T1 and T2) tumors as well as patients without neck lymph node metastasis and in I and II clinical stages showed a significantly improved disease-specific survival (DSS) (Appendix A) (*p* = 0.001, *p* = 0.01, and *p* = 0.02, respectively). Stromal infiltrating CD20^+^ TILs did not show a significant relationship with survival. However, a significant relationship was detected between the low density of tumoral CD20^+^ B cells and a lower DSS (*p* = 0.04) (Figure 3).

The different ratios between stromal/tumoral CD20^+^ and the other TIL and macrophage markers did not show any significant association with survival (Appendix A).

We also performed a stratified univariate Kaplan–Meier analysis according to tumoral CD20 TIL infiltration (Appendix A). Low density of tumoral CD20^+^ B cells was significantly associated with a reduced DSS in patients younger than 65 years (*p* = 0.03), male patients (*p* = 0.02), and in cases with tobacco (*p* = 0.004), or alcohol consumption (*p* = 0.005).

Our data showed that a low density of tumoral CD20^+^ TILs were associated with a poorer DSS in patients with positive tumoral PD-L1 expression (*p* = 0.008), negative SOX2 (*p* = 0.01) or negative NANOG expression (*p* = 0.03), in cases with high infiltration of CD8^+^ TILs in the tumor nests (*p* = 0.03), and in cases with low density of stromal CD4^+^ and tumoral CD68^+^ cells (*p* = 0.03) (Appendix A). Finally, on multivariate analysis, clinical stage (stages I–II vs. III–IV), and tumoral CD20^+^ infiltrating cells (low vs. high density) were the only parameters independently associated with DSS (HR = 2.42, *p* = 0.003; HR = 0.57, *p* = 0.04, respectively).

## 3. Discussion

Lymphocyte infiltration into the TME is generally considered to represent host immunity against tumors [35]. To date, most studies have evaluated the relevance of infiltrating T cells in the TME, while less attention has been focused on the significance of B cells in OSCC with conflicting results [7,36,37]. Nonetheless, there is increasing evidence that infiltrating B cells may influence the TME towards tumorigenic or cytotoxic [38]. In tumors such as cutaneous melanoma, tumor-infiltrating B cells have been positively associated with patient prognosis in some studies [39,40], whereas this association was demonstrated in others [5], and even a negative association has also been reported [41]. Here, we found that CD20^+^ TIL infiltration in the tumor, but not the stroma, is associated with a better outcome. This finding is in agreement with other reports [11,15,33,35,42,43], suggesting that an immune response may be mediated by B lymphocytes. Furthermore, our study revealed that CD20^+^ TIL density in the tumor nests was an independent prognostic factor in this OSCC patient cohort. A stratified survival analysis showed that tumoral CD20^+^ TILs were significantly associated with a better prognosis in patients younger than 65 years, male patients, with tobacco or alcohol consumption, positive PD-L1 expression in more than 10% of tumor cells, tumors with negative expression of SOX2 or NANOG, high density of tumoral CD8^+^ TILs, low density of tumoral infiltration by CD68^+^ macrophages and low stromal infiltration by CD4^+^ TILs. Nielsen et al. [44] found that CD20^+^ B cells co-localized with CD8^+^ T cells in high-grade serous ovarian cancer, raising the possibility that B lymphocytes may act as antigen-presenting cells to facilitate the antitumoral T cell cytolytic response. Several mechanisms can explain the divergent roles that B cells play in tumor immunology comparable to a double-edged sword. On one hand, tumor-infiltrating B cells can secrete lymphotoxin, which induces angiogenesis, and activates NF-κB signaling and STAT3 in the cancer cells thereby promoting tumor growth [3]. Furthermore, extracellular vesicles derived from tumors are capable of activating B cells to produce antibodies which, in turn, can form immune complexes [45], activating F_Cγ_ receptors on myeloid cells, and suppressing antitumor CD4^+^ and CD8^+^ T cell responses [3]. On the other hand, antibodies against tumor-specific antigens produced by plasma cells have different roles: antibodies mediate complement-dependent tumor cell lysis [44], Fc-mediated phagocytosis by macrophages, as well as antibody-dependent cellular cytotoxicity by natural killer (NK) cells [3,46]. In addition, antibody-coated tumor cells could also be processed by dendritic cells, which in turn present tumor antigens to CD4^+^ T cells and cross-present antigens to CD8^+^ T cells, aiding in the immune response against tumor cells [33,47]. Finally, lymphotoxin produced by B cells has an additional role in promoting the formation of ectopic tertiary lymphoid organs, which correlate directly with a positive outcome in many cancers [3,48,49]. B cells may concentrate on TME or form tumor-associated immune aggregates that include tertiary lymphoid structures (TLS), similar to lymph nodes [50]. TLS can be localized around the tumor or even within the tumor itself, showing different stages of maturation defined by the absence or presence of one or more germinal centers surrounded by T cells, dendritic and plasma cells, along with lymphatic and blood vessels [51]. B cells could be involved in the formation of TLS by producing CXCL13 and lymphotoxin [50,52], and also in the maturation of an antitumoral humoral response in the TME [50]. TLS have been reported in various types of cancers including lung, breast, pancreas, colorectal, and oral cancer [53,54]. These structures have been associated with positive prognostic value in some tumors [55,56], such as high-grade serous ovarian cancer, where tumor infiltration by CD8^+^ T cells only showed prognostic value when it was combined with the presence of TLS and a high count of plasma cells, CD4^+^ T cells and CD20^+^ B cells [56]. In oral cancer, higher grades of TLS have also been associated with improved survival [57]; however, good and poor clinical outcomes have been correlated with different cellular components of TLS, such as dendritic cells, B cells, and different subsets of T cells [58]. This suggests that the cellular components of TLS affect the antitumor immune response in different cancer types, such as gastric cancer, where a high number of CD20^+^ B cells within lymphoid aggregates was an independent predictor of good prognosis [59]. Moreover, B cells are not only cellular precursors of antibody-producing plasma cells in the TME, but they also act as antigen-presenting cells and are also capable of directly killing cancer cells through the release of granzyme B [60]. These functions support a tumor-suppressive role for B cells; however, immunosuppressive regulatory B cells that produce TGF-β and IL-10, promoting the same effects as Treg cells, have also been found [61]. This heterogeneity in B cell function may explain the contradictory results among studies where pan B cell markers are used. B cells may inhibit the immune response against tumors, but the underlying mechanism is poorly understood. However, it is well-known that B cells expressing PD-L1 interact with follicular T helper (Tfh) cells with high expression of the programmed cell death-1 (PD-1) molecule, thus suppressing CXCR3 upregulation and the follicular recruitment of activated helper T cells [62].

Even though intratumoral CD20^+^ TIL density was independently associated with the prognosis in this OSCC patient cohort, other important prognostic clinicopathological variables, such as age, tumor stage, neck lymph node metastasis, histological grade, or tumor location within the oral cavity were not associated with CD20^+^ TIL infiltration. Pretscher et al. [35] found that intratumoral CD20^+^ B cells were increased in number in metastasis compared to primary tumors, while Taghavi et al. [42] observed a significant inverse association between peritumoral CD20^+^ B cells and lymph node metastasis. We did not find any association between CD20^+^ TILs and metastasis or clinical stage. Noteworthy, we found a significant reduction in the number of infiltrating CD20^+^ B cells in T3 and T4 tumors compared with smaller cancer sizes, which suggests that the reduction in the tumoral B lymphocyte infiltration could be associated with tumor progression. Conversely, it is also plausible that the upfront presence of fewer immune cells and therewith inadequate immunosurveillance of the tumor could ultimately favor tumor growth. In contrast to the above results, other studies have demonstrated an increase in CD20^+^ B cell infiltration in association with a poorer prognosis [7,8]. The conflicting results may be due to differences in immunohistochemical procedures and/or scoring used.

CSCs can attract macrophages into the tumors [63] and induce the M2 phenotype, secreting IL-6, IL-10, TGF-β, and EGF and driving CSC self-renewal by activating the STAT3/NF-κB signaling pathway [64]. Complementarily exosomes derived from tumors, including OSCC, and released by several types of cells, including B lymphocytes, can activate M2 TAMs [65]. The cross-talk between CSCs and TAMs is orchestrated by the STAT3 signaling pathway [66], which promotes stemness, survival, and proliferation in CSCs. Conversely, CSCs can induce the immunosuppressive properties of TAMs repressing T lymphocytes [64]. We found a significant and inverse association between a higher tumoral infiltration of CD20^+^ B cells with negative SOX2 expression. Similarly, CD163^+^ TAM infiltration in OSCC has also been inversely correlated with the expression of the CSC markers NANOG and SOX2 [67] Together these data suggest an inverse relationship between B-cell infiltration and stemness in OSCC, which could plausibly reflect CSCs role in immune evasion and the contribution to OSCC progression. To the best of our knowledge, this is the first study to provide a potential link between CD20^+^ B lymphocytes and CSCs.

Regarding the TME, both inflamed and non-inflamed phenotypes have been described [11]. OSCCs have mostly inflamed phenotypes [17], and it has been generally thought that, the more abundant the inflammatory infiltration surrounding the tumor nests, the better the prognosis of the patients. However, inflammation in OSCCs has been generally associated with poor survival [6]. Necrosed tumor cells may contribute to an inflamed TME and also proinflammatory cytokines released by CSCs, such as IL-6, IL-8, IL-10, and IL-13 can contribute to maintaining an inflammatory and suppressive TME representing the “niche” sustaining cellular stemness [68].

## 4. Materials and Methods

### 4.1. Patients and Tissue Specimens

A cohort of 125 patients with histologically confirmed OSCC surgically treated at the Hospital Universitario Central de Asturias between 1996 and 2007 was retrospectively selected for the study. This 125 OSCC cohort was previously described [24,25], predominantly male patients (*n* = 82, 65.6%), ranging in age from 28 to 91 years (mean 58.69, standard deviation 14.34 years). Eighty-four (67%) patients were active smokers and 69 (55%) alcohol drinkers. Primary tumors were located in the mobile tongue (*n* = 51, 41%), floor of the mouth (*n* = 37; 30%), gingiva (*n* = 22; 18%), buccal mucosa (*n* = 7; 6%), retromolar area (*n* = 6; 4%), and palate (*n* = 2; 1%). The patients were staged according to the 8th edition of the TNM classification of malignant tumors [69]. Regarding pT, 27 (22%) cases were classified as T1, 54 (43%) T2, 16 (13%) T3, and 28 (22%) T4. 49 (39%) patients presented lymph node metastasis (pN+), whereas the remaining 76 (61%) patients showed an absence of neck lymph node metastasis (pN0). The distribution according to overall AJCC stages was as follows: 20 (16%) patients were stage I, 32 (26%) patients were stage II, 26 (20%) patients were stage III, and finally, 47 (38%) patients were stage IV. Regarding histopathologic degree of differentiation, 80 (64%) OSCCs were well-differentiated, 41 (33%) moderately, and 4 (3%) poorly-differentiated. None of the patients underwent chemotherapy or radiotherapy before surgical treatment. Complementary radiotherapy (*n* = 75; 60%), or chemotherapy (*n* = 14; 11%) were administered when indicated.

53 (44%) patients were alive and free of recurrence during the follow-up period (6 to 230 months), while 19 (15%) suffered from a second primary cancer in the oral cavity, and 51 (42%) died of OSCC or showed a non-treatable recurrence. Disease-specific survival (DSS) was the clinical endpoint, calculated as the time interval from the initial surgical treatment to the date of death by the tumor or the presence of a non-treatable recurrence.

### 4.2. Immunohistochemistry (IHC)

Tissue microarrays (TMAs) were constructed by collecting three morphologically representative areas (1 mm diameter) from each formalin-fixed, paraffin-embedded (FFPE) tumor block, as previously reported [24,25]. Each TMA contained morphologically normal oral mucosa samples from non-oncological patients undergoing oral surgery, used as internal controls. The TMAs were cut into 3-μm sections and dried on Flex IHC microscope slides (DakoCytomation, Glostrup, Denmark). Tissue sections were deparaffinized in xylene and rehydrated in serial baths of ethanol, and then blocked with 3% hydrogen peroxide. Antigen retrieval was done using Envision Flex Target Retrieval solution, high pH (Dako), followed by staining on an automatic staining workstation (Dako Autostainer Plus, Dako). Immunohistochemistry was performed incubating the slides with monoclonal antibodies against CD20 (Dako, clone L26, catalogue number M0755; 1:200 dilution), CD4 (Dako, clone 4B12, 1:80 dilution), CD8 (Dako, clone C8/144B, prediluted), FoxP3 (Cell Signaling Technology, clone D6O8R, 1:100 dilution), CD68 (Agilent-Dako, clone KP1, prediluted), CD163 (Biocare Medical, clone 10D6, 1:100 dilution), PD-L1 antibody (clone 22C3; 1:200 dilution; PD-L1 IHC 22C3 pharmDx; Dako SK006), NANOG (D73G4 XP^®^; 1:200 dilution, Cell Signaling Technology, Inc.), and SOX2 (AB5603; 1:1000 dilution, Merck Millipore). The antibody-antigen complex was visualized with the Dako EnVision Flex + Visualization System (Dako). Tonsil tissue was used as a positive control for CD20.

The IHC results were independently evaluated by four observers (FDI, JSC, JPR, and JMG-P), blinded to clinical information. The number of CD20^+^, CD68^+^, CD163^+^, CD4^+^, CD8^+^, and FOXP3^+^ cells was counted in each 1 mm^2^ area from three independent high-power representative microscopic fields (HPFs, 400×; 0.0625 μm^2^), both in the tumor nests and tumor stroma. The median was used as a cut-off to separate patient groups based on these lymphocyte markers. Thus, CD20^+^, CD4^+^, CD8^+^, CD68^+^, CD163^+^, and FOXP3^+^ cells immunostaining was classified into two groups, above and below median number of staining for the total patient population. Stromal and tumoral ratios between CD20 and the remaining immune cell markers (CD8, CD4, FOXP3, CD68, and CD163) were calculated. PD-L1 expression in more than 10% of tumor cells was significantly associated with poorer survival in a previous study [34], and accordingly established as a cut-off point for subsequent analyses. SOX2 expression was evaluated as the percentage of tumor cells with positively stained nuclei, as previously described [24,25]. SOX2 staining scores were classified as negative or positive expression as below or above the median cut-off value of 10%, respectively. NANOG staining intensity was scored as negative (score 0) *versus* positive expression (scores 1–2).

### 4.3. Statistical Analysis

Statistical analyses were carried out using SPSS software version 18 (IBM Co., Armonk, NY, USA). Continuous variables (CD20^+^, CD68^+^, CD163^+^, CD4^+^, CD8^+^, and FOXP3^+^ cells) were expressed as the means ± standard deviations (SD), and absolute and relative frequencies were calculated for categorical variables. The correlations between the numbers of stromal or tumoral CD20^+^, and CD68^+^, CD163^+^, CD4^+^, CD8^+^, and FOXP3^+^ cells were assessed using Spearman’s rank correlation coefficient. Mann–Whitney *U* test was used to evaluate the relationship between CD20 expression and the different clinicopathological variables. Analyses of disease-specific survival (DSS) were performed using the Kaplan–Meier method, and a comparison of survival rates was performed by using the log-rank test. Moreover, the univariate and multivariate Cox regression model was applied to calculate hazard ratios (HRs) and 95% confidence intervals (95% CI), as well as to determine independent prognostic factors in the presence of other prognostically relevant covariates. All *p*-values were based on two-sided statistical analysis, and for all analyses a *p*-value, less than 0.05 was considered to be statistically significant.

## 5. Conclusions

This study thoroughly investigated the clinical relevance of B cell infiltration in the context of oral TME complexity, by jointly evaluating both in the tumor nests and surrounding stroma associations between infiltrating CD20^+^ B lymphocytes and other immune profiles CD4^+^, CD8^+^, and FOXP3^+^ TILs, CD68^+^ and CD163^+^ macrophages in OSCC specimens. Our findings demonstrate that B cell infiltration had an impact on OSCC patient prognosis. Interestingly, high CD20^+^ TIL density in the tumor nests emerges as an independent good prognostic factor in OSCC, thus underlining a potential role of B lymphocytes in antitumor immunity in oral cancers. Furthermore, this study also uncovered an inverse correlation between infiltration of CD20^+^ B cells and the expression of CSC markers, in particular SOX2. Nevertheless, these results need further confirmation using large independent validation cohorts of OSCC and/or other HNSCC patients.

## Figures and Tables

**Figure 1 cancers-13-00395-f001:**
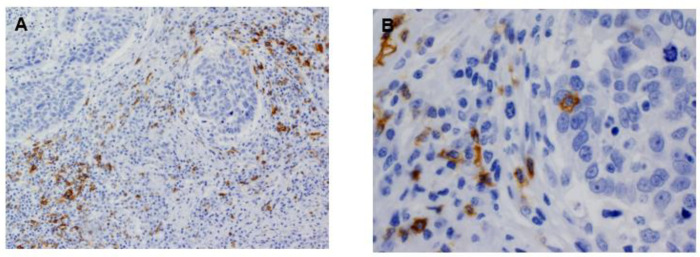
Immunohistochemical analysis of CD20^+^ B lymphocytes in oral squamous cell carcinoma (OSCC) tissue specimens. (Original magnification × 200 in panel (**A**), × 400 Figure in panel (**B**)).

**Figure 2 cancers-13-00395-f002:**
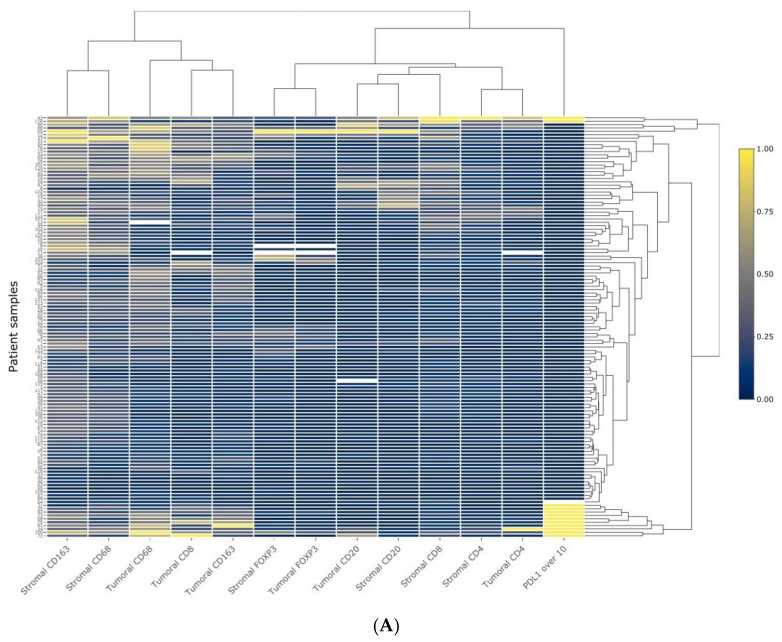
Heatmap and hierarchical clustering analysis to classify the different variables studied. (**A**) Heatmap displaying the IHC scores for the different immune profiles in both tumoral and stromal compartments. (**B**) Heatmap displaying the clinical characteristics, IHC scores for CD20^+^ B cell infiltration, Nanog, and SOX2 expression. All data sets were normalized.

**Figure 3 cancers-13-00395-f003:**
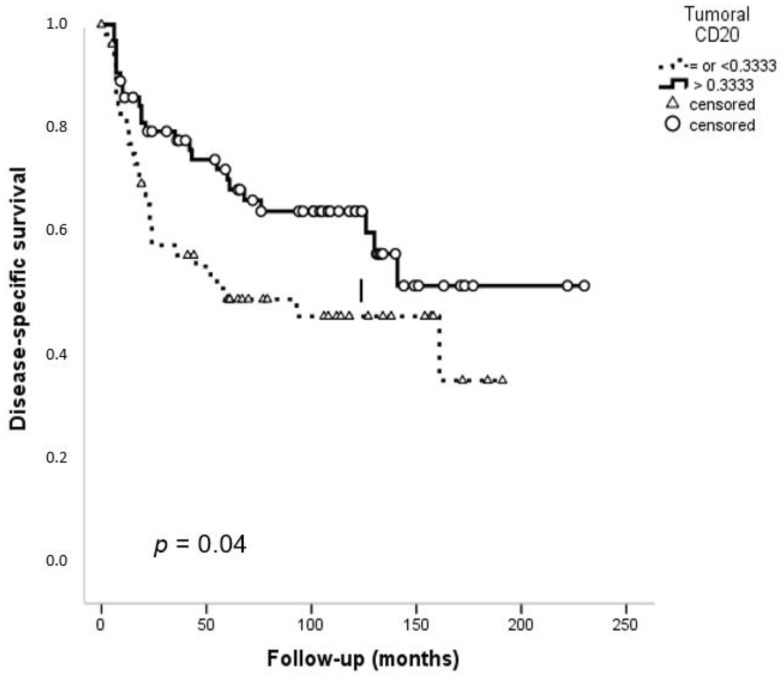
Kaplan–Meier disease-specific survival curves in the cohort of 125 OSCC patients categorized by tumoral CD20^+^ B cell infiltration. Median values were used as cut-off points. *p* values were estimated using the log-rank test.

**Table 1 cancers-13-00395-t001:** Correlations between the mean numbers of CD20^+^ B cells with CD4^+^, CD8^+^, and FOXP3^+^ TILs, and CD68^+^ and CD163^+^ macrophages infiltrating both the tumor nests and surrounding stroma. The Spearman’s Rho coefficients and the corresponding *p* values are shown.

Variable	Stromal CD20^+^ (Mean)	Tumoral CD20^+^ (Mean)
Stromal CD8^+^ (mean)	0.772 <0.001	0.624 <0.001
Tumoral CD8^+^ (mean)	0.404 <0.001	0.358 <0.001
Stromal CD4^+^ (mean)	0.662 <0.001	0.488 <0.001
Tumoral CD4^+^ (mean)	0.353 <0.001	0.362 <0.001
Stromal FOXP3^+^ (mean)	0.326 <0.001	0.328 <0.001
Tumoral FOXP3^+^ (mean)	0.253 0.005	0.246 0.006
Stromal CD68^+^ (mean)	0.354 <0.001	0.377 <0.001
Tumoral CD68^+^ (mean)	0.259 0.004	0.261 0.004
Stromal CD163^+^ (mean)	0.418 <0.001	0.373 <0.001
Tumoral CD163^+^ (mean)	0.190 0.03	0.228 0.01

**Table 2 cancers-13-00395-t002:** Associations between stromal/tumoral CD20^+^ B infiltration and the clinicopathological parameters in the cohort of 125 OSCC patients.

Variable	Number	Stromal CD20^+^ Mean (SD)	*p*	Tumoral CD20^+^ Mean (SD)	*p*
Age (years)					
<65	77	35.06 (74.34)	0.35	1.26 (2.90)	0.14
≥65	48	54.36 (84.50)		2.21 (3.92)	
Gender					
Female	43	46.61 (82.46)	0.81	2.42 (4.31)	0.27
Male	82	40.30 (76.97)		1.22 (2.66)	
Tobacco consumption					
No	41	48.43 (77.72)	0.25	2.40 (3.99)	0.06
Yes	84	39.56 (79.37)		1.26 (2.95)	
Alcohol consumption					
No	56	42.01 (74.87)	0.90	2.14 (3.88)	0.09
Yes	69	42.84 (82.09)		1.11 (2.74)	
pT classification					
T1 + T2	81	48.91 (83.98)	0.02	1.95 (3.76)	0.10
T3 + T4	44	30.62 (66.97)		1.02 (2.29)	
pN classification					
N0	76	41.03 (74.05)	0.63	1.73 (3.42)	0.70
N+	49	44.70 (86.00)		1.47 (3.26)	
Stage					
I + II	52	51.18 (83.82)	0.31	2.22 (3.90)	0.15
III + IV	73	36.26 (74.68)		1.20 (2.84)	0.15
Grade					
Well	80	52.93 (91.88)	0.06	1.99 (3.94)	0.31
Moderate + Poor	45	23.88 (41.41)		1.00 (1.78)	
Site					
Tongue	51	57.11 (103.48)	0.37	2.09 (4.26)	0.74
Other	74	32.38 (53.99)		1.30 (2.51)	
Recurrence					
No	71	39.68 (76.77)	0.36	1.44 (3.32)	0.90
Yes	54	46.14 (81.59)		1.87 (3.39)	
Second primary tumor					
No	106	37.61 (76.86)	0.02	1.30 (2.71)	0.24
Yes	19	69.56 (84.97)		3.45 (5.46)	

All *p*-values were calculated using the Mann–Whitney U test.

**Table 3 cancers-13-00395-t003:** Associations between stromal and tumoral CD20/CD8, CD20/CD4, and CD20/FOXP3 ratios and clinicopathological parameters in the cohort of 125 OSCC patients.

Variable Mean (SD)	Stromal CD20/CD8 Ratio	*p*	Tumoral CD20/CD8 Ratio	*p*	Stromal CD20/CD4 Ratio	*p*	Tumoral CD20/CD4 Ratio	*p*	Stromal CD20/FOXP3 Ratio	*p*	Tumoral CD20/FOXP3 Ratio	*p*
Age (years)												
<65	0.50 (3.06)	0.50	0.04 (0.10)	0.35	0.71 (1.10)	0.88	1.10 (3.97)	0.92	3.50 (8.34)	0.01	0.40 (0.75)	0.01
≥65	0.20 (0.24)		0.21 (0.60)		0.78 (1.18)		0.73 (2.25)		16.38 (42.95)		1.99 (4.30)	
Gender												
Female	0.16 (0.18)	0.88	0.08 (0.23)	0.46	0.66 (1.07)	0.61	0.69 (2.30)	0.91	14.03 (45.37)	0.76	2.05 (4.57)	0.55
Male	0.50 (2.97)		0.13 (0.47)		0.78 (1.16)		1.08 (3.79)		5.88 (12.30)		0.46 (0.68)	
Tobacco												
No	0.17 (0.16)	0.47	0.07 (0.23)	0.50	0.69 (0.96)	0.68	0.93 (2.51)	0.57	17.25 (45.85)	0.16	1.62 (4.12)	0.20
Yes	0.48 (2.93)		0.13 (0.46)		0.76 (1.20)		0.94 (3.70)		4.34 (11.08)		0.68 (1.72)	
Alcohol												
No	0.15 (0.16)	0.81	0.12 (0.44)	0.32	0.62 (1.00)	0.50	0.65 (2.06)	0.92	12.71 (39.44)	0.46	1.50 (3.64)	0.06
Yes	0.57 (3.23)		0.10 (0.36)		0.83 (1.21)		1.22 (4.19)		5.16 (12.28)		0.60 (1.80)	
pT												
T1 + T2	0.52 (2.98)	0.03	0.15 (0.48)	0.07	0.86 (1.26)	0.13	0.87 (2.27)	0.15	7.60 (15.79)	0.21	1.25 (3.27)	0.75
T3 + T4	0.12 (0.15)		0.03 (0.09)		0.51 (0.79)		1.08 (4.78)		11.02 (46.03)		0.40 (0.65)	
pN												
N0	0.51 (3.08)	0.47	0.07 (0.29)	0.41	0.66 (1.02)	0.53	0.62 (1.88)	0.69	9.75 (34.76)	0.29	1.25 (3.42)	0.82
N+	0.18 (0.23)		0.17 (0.52)		0.85 (1.28)		1.42 (4.70)		7.10 (14.57)		0.56 (0.86)	
Stage												
I + II	0.70 (3.72)	0.35	0.11 (0.35)	0.93	0.76 (1.15)	0.77	0.78 (2.23)	0.14	6.86 (14.90)	0.94	1.61 (3.93)	0.64
III + IV	0.15 (0.20)		0.11 (0.43)		0.72 (1.11)		1.04 (3.86)		10.15 (35.83)		0.46 (0.75)	
Grade												
Well	0.52 (3.00)	0.29	0.08 (0.28)	0.22	0.78 (1.14)	0.19	0.73 (1.97)	0.19	11.36 (34.96)	0.09	1.35 (3.54)	0.69
Moderate + Poor	0.14 (0.17)		0.16 (0.55)		0.67 (1.10)		1.27 (4.72)		4.20 (10.36)		0.47 (0.56)	
Site												
Tongue	0.71 (3.76)	0.23	0.08 (0.28)	0.64	0.73 (1.17)	0.74	1.11 (4.39)	0.33	4.68 (10.99)	0.93	0.94 (2.21)	0.29
Other	0.15 (0.20)		0.13 (0.47)		0.75 (1.10)		0.79 (2.07)		11.38 (35.62)		1.04 (3.15)	
Recurrence												
No	0.16 (0.22)	0.46	0.14 (0.49)	0.91	0.74 (1.15)	0.64	0.53 (1.60)	0.90	6.69 (16.84)	0.01	0.66 (1.86)	0.37
Yes	0.67 (3.65)		0.07 (0.21)		0.73 (1.10)		1.43 (4.58)		11.59 (39.79)		1.56 (3.83)	
Second primary tumor												
No	0.40 (2.61)	0.01	0.12 (0.43)	0.38	0.66 (1.08)	0.02	0.99 (3.58)	0.23	7.88 (29.81)	0.10	0.88 (2.65)	0.57
Yes	0.26 (0.21)		0.07 (0.13)		1.16 (1.28)		0.68 (1.01)		12.87 (22.13)		1.58 (3.42)	

All *p*-values were calculated using the Mann–Whitney U test.

**Table 4 cancers-13-00395-t004:** Association between CD20^+^ TILs, PD-L1, and CSCs markers.

Factor	Stromal CD20^+^ (Mean, SD)	*p*	Tumoral CD20^+^ (Mean, SD)	*p*
SOX2				
Negative (*N* = 72, 60%)	48.47 (77.74)	0.29	1.86 (3.38)	0.008
Positive (*N* = 49, 40%)	36.23 (82.77)		1.31 (3.39)	
NANOG				
Negative (*N* = 83, 68%)	44.10 (77.05)	0.14	1.86 (3.67)	0.36
Positive (*N* = 39, 32%)	40.23 (85.20)		1.23 (2.65)	
PD-L1				
≤10% (*N* = 104, 83%)	43.96 (79.55)	0.85	1.46 (3.30)	0.21
>10% (*N* = 18, 15%)	39.27 (80.81)		2.70 (3.67)	

## Data Availability

The data presented in this study are available within the article and Appendix A.

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
