# Peer review of "Tumor-Infiltrating CD20+ B Lymphocytes: Significance and Prognostic Implications in Oral Cancer Microenvironment"

_cancers, 2021, doi:10.3390/cancers13030395_

Round 1

Reviewer 1 Report

The authors studied the clinical impact of tumor-infiltrating CD20+ B lymphocytes over tumor specimen in the micro-environment from paraffin blocks in the oral cancer. This is a well written manuscript but there are some concerns listed below.

  1. Please use the tumor staging system according to 8th edition of AJCC in this study although this cohort is enrolled in earlier date.
  2. It is time consuming to check the IHC staining in the tumor micro-environment. The B cell or other immune cells infiltration may be non-homogeneous distribution over the tumor block. How did the authors choose the hot area of tumor for study?
  3. This bias may exist by visual inspection for these IHC staining by pathologists. Perhaps, the application of digital analysis is more reliable and precise.

Author Response

Reviewer #1

Comments and Suggestions for Authors

The authors studied the clinical impact of tumor-infiltrating CD20+ B lymphocytes over tumor specimen in the micro-environment from paraffin blocks in the oral cancer. This is a well written manuscript but there are some concerns listed below.

Point 1: Please use the tumor staging system according to 8th edition of AJCC in this study although this cohort is enrolled in earlier date.

Response 1: We thank the reviewer for this observation. We must clarify that the 8th edition of AJCC was actually used for tumor staging in our study cohort; however, the reference was incorrect. This has been amended in the new version.

Lydiatt, W.M.; Patel, S.G.; Ridge, J.A.; O’Sullivan, B.; Shah, J.P. Staging head and neck cancers. In: AJCC Cancer Staging Manual. 8th ed.; Chicago, IL, USA, 2017, pp. 55 - 181.

Point 2: It is time consuming to check the IHC staining in the tumor micro-environment. The B cell or other immune cells infiltration may be non-homogeneous distribution over the tumor block. How did the authors choose the hot area of tumor for study?

Response 2: Scoring of IHC staining in the tumor microenvironment was simultaneously performed by three pathologists, reaching an agreement on the final figure. In fact, the possible non-homogeneous distribution of tumor-infiltrating immune cells was considered, thereby taking 3 representative areas per tumor block, including both the center of the tumor and the invasive front. No significant differences were found between the 3 tumor areas assessed and staining scores were averaged to obtain a final score. The differential distribution of immune subpopulations in both the malignant tumoral nests the surrounding stroma was only determined.

Point 3: This bias may exist by visual inspection for these IHC staining by pathologists. Perhaps, the application of digital analysis is more reliable and precise.

Response 3: Certainly, visual inspection of IHC staining is time-consuming and may be subjected to observation bias, while digital analysis could be a priori a more precise and objective procedure. However, we truly believe that visual analysis provides an important added value to this particular study, as it was fundamental to carefully discriminate between stroma and tumor, and to precisely score staining for each immune cell subtype and compartment. In particular, this was critical for the assessment of the macrophage markers CD68 and CD163, which can also mark other myeloid cells, including neutrophils, as reported in the literature.

Reviewer 2 Report

The authors report on the analysis of stromal and tumoral CD20+ B lymphocytes in 125 OSCC patients and found that, although infiltration of both stromal and tumoral CD20+ TILs were strongly correlated  with other immune profiles (i.e. CD4+, CD8+ and FOXP3+ TILS, and CD68+ and CD163+ macrophages), only tumoral CD20+ TILs were independently associated with disease-specific survival. The reported findings are very interesting and of relevance since the prognostic significance of B-cells in tumor immunology is upcoming, however several short comings need to be addressed.

Most importantly, the obtained data provided by this research is very interesting, but the way these results were analyzed and presented do not fully justify the data, sometimes lacking depth. To increase the article's depth the authors should consider adding additional analysis, e.g. hierarchical clustering analysis. A heatmap displaying the infiltration of the different immune profiles in the tumoral and stromal compartments and clinical characteristics and/or a heatmap displaying the immune profiles and analysis of the NANOG and SOX2 expression would be of additive value.

Furthermore, statistical testing for determining disease-specific-survival was performed using univariate cox regression analysis (table 5). Since age, TNM status and alcohol/tobacco use are basic prognostic factors, a multivariable Cox regression analysis should be added. In addition, to enhance ease of reading and quick interpretation of the results, extensive tables should be avoided and included as supplement. Instead, additional figures, e.g. forest plots displaying the hazard ratios, should be added.

Finally, although tertiary lymphoid structures (TLS) are often not captured inside a Tissue Micro Array (TMA), the possible role of tumoral and stromal B-cells can hardly be described outside the context of TLSs. Only one non-representative sentence about TLSs has been included in the manuscript (line 237: “Finally, lymphotoxin produced by B cells has an additional role promoting the formation of ectopic tertiary lymphoid organs, which correlate directly with positive outcome in many cancers”), while this deserves a small section on its own. 

Other points of improvement are:

  • Please specify whether specimens from the base of the tongue were included or excluded since this site is naturally rich in lymphatic tissue.

  • Viral agents that are linked to human tumours, such as human papillomavirus (HPV), has become a great topic of interest during the last years. While HPV probably plays a minor role in cancers arising in the oral cavity it would still be interesting and of additive value to evaluate HPV-status and relate this to B-cell infiltration and other immune profiles.

  • Table 3:
    number of patients is not corresponding with text (page 6, line 136-147).

  • Table 4:
    Percentages in text do not correspond with percentages in table.
  • Page 19, line 217: “a stratified survival analyses showed that tumoral CD20+ TILs were significantly associated with prognosis in […]”

Please specify high or low density of tumoral CD20+ TILs.

  • Page 20, line 254-257: “Noteworthy, we found a significantly reduction in the number of infiltrating CD20+ B cells in T3 and T4 tumors compared with smaller cancer sizes, which suggests that the reduction in the tumoral B lymphocyte infiltration could be associated with tumor progression”.
  • It could be the other way around; due to the upfront presence of less immune cells and therewith inadequate immunosurveillance of the tumor, the tumor has the opportunity to grow.

Typing/grammatical errors:

  • Page 3, line 10: “[…] as well as tumor PD-L1 and two important CSCs-related factors NANOG and SOX2”.
  • PD-L1 expression by tumor cells, or tumoral PD-L1 expression

  • Page 6, line 134: “it was no possible to calculate the ratios of infiltrating CD20+ B cells.

  • Page 6 line 141 and 146: “being this ratio”

  • Page 20, line 294: “Eight-four (67%) patients were active smokers, and 69 (55%) were”

Author Response

Reviewer #2

Comments and Suggestions for Authors

Point 1: The authors report on the analysis of stromal and tumoral CD20+ B lymphocytes in 125 OSCC patients and found that, although infiltration of both stromal and tumoral CD20+ TILs were strongly correlated  with other immune profiles (i.e. CD4+, CD8+ and FOXP3+ TILS, and CD68+ and CD163+ macrophages), only tumoral CD20+ TILs were independently associated with disease-specific survival. The reported findings are very interesting and of relevance since the prognostic significance of B-cells in tumor immunology is upcoming, however several short comings need to be addressed. Most importantly, the obtained data provided by this research is very interesting, but the way these results were analyzed and presented do not fully justify the data, sometimes lacking depth. To increase the article's depth the authors should consider adding additional analysis, e.g. hierarchical clustering analysis. A heatmap displaying the infiltration of the different immune profiles in the tumoral and stromal compartments and clinical characteristics and/or a heatmap displaying the immune profiles and analysis of the NANOG and SOX2 expression would be of additive value.

Response 1: We thank the reviewer for his/her positive comments highlighting the interest and relevance of our findings. We also appreciate the meticulous revision and insightful suggestions.

Following the reviewer’s recommendation, we performed hierarchical clustering analysis of stromal and tumoral B-cell infiltration with other immune subtypes (lymphocyte and macrophage markers), and also with clinicopathological variables, NANOG and SOX2 expression. These data have been included in new Figure 2, shown as heatmaps with dendrograms displaying the closeness of clusters.

Point 2: Furthermore, statistical testing for determining disease-specific-survival was performed using univariate cox regression analysis (table 5). Since age, TNM status and alcohol/tobacco use are basic prognostic factors, a multivariable Cox regression analysis should be added. In addition, to enhance ease of reading and quick interpretation of the results, extensive tables should be avoided and included as supplement. Instead, additional figures, e.g. forest plots displaying the hazard ratios, should be added.

Response 2: In addition to the univariate Cox analysis shown in Table 5 we also performed multivariate Cox analysis, as mentioned at the end of Results section (lines 242-244). However, since age, gender, tobacco and alcohol consumption did not reach statistically significance, for this reason these variables were not further included in the multivariate analysis. Nevertheless, following the reviewer’s suggestion, we have repeated the multivariate Cox regression analysis now including age, gender, tobacco and alcohol, with the following results:

Variable

P

Hazard Ratio

Age

0.21

1.49

Gender

0.63

0.85

Tobacco

0.34

1.53

Alcohol

0.29

0.64

TNM stage

0.004

2.46

Stromal CD20

0.29

1.36

Tumoral CD20

0.01

0.48

As in our previous multivariate analysis, disease stage and tumoral CD20+ B cell infiltration were confirmed as the only parameters independently associated with DSS.

Point 3: Finally, although tertiary lymphoid structures (TLS) are often not captured inside a Tissue Micro Array (TMA), the possible role of tumoral and stromal B-cells can hardly be described outside the context of TLSs. Only one non-representative sentence about TLSs has been included in the manuscript (line 237: “Finally, lymphotoxin produced by B cells has an additional role promoting the formation of ectopic tertiary lymphoid organs, which correlate directly with positive outcome in many cancers”), while this deserves a small section on its own.

Response 3: We agree. This information has been further extended in the Discussion (lines 278-287), as it follows: B cells may concentrate in TME or form tumor-associated immune aggregates that include tertiary lymphoid structures (TLS), similar to lymph nodes [1]. TLS can be localized around the tumor or even within the tumor itself, and they have one or more germinal centers surrounded by T cells, dendritic and plasma cells, along with lymphatic and blood vessels [2]. B cells could be involved in the formation of TLS by producing CXCL13 and lymphotoxin [1, 3], and also in the maturation of an antitumoral humoral response in the TME [1]. These structures have been associated with positive prognostic value in some tumors [2, 4], such as high-grade serous ovarian cancer, where tumor infiltration by CD8+ T cells only showed prognostic value when it was combined with the presence of TLS and a high count of plasma cells, CD4+ T cells and CD20+ B cells [5].

  1. Sharonov, G.V.; Serebrovskaya, E.O.; Yuzhakova, D.V.; Britanova, O.V.; Chudakov, D.M. B cells, plasma cells and antibody repertoires in the tumour microenvironment. Rev. Immunol. 2020,20, 294-307. https://doi: 10.1038/s41577-019-0257-x.
  2. Sautès-Fridman, C.; Petitprez, F.; Calderaro, J.; Fridman, W.H. Tertiary lymphoid structures in the era of cancer immunotherapy. Rev. Cancer. 2019,19, 307-325. https://doi: 10.1038/s41568-019-0144-6.
  3. Siliņa, K.; Rulle, U.; Kalniņa, Z.; Linē, A. Manipulation of tumour-infiltrating B cells and tertiary lymphoid structures: a novel anti-cancer treatment avenue? Cancer Immunol. Immunother. 2014, 63, 643-62. https://doi: 10.1007/s00262-014-1544-9.
  4. Colbeck, E.J.; Ager, A.; Gallimore, A.; Jones, G.W. Tertiary lymphoid structures in cancer: Drivers of antitumor immunity, immunosuppression, or bystander sentinels in disease? Immunol. 2017,9, 1830. https://doi: 10.3389/fimmu.2017.01830.
  5. Kroeger, D.R.; Milne, K.; Nelson, B.H. Tumor-infiltrating plasma cells are associated with tertiary lymphoid structures, cytolytic T-cell responses, and superior prognosis in ovarian cancer. Cancer Res. 2016,22, 3005-3015. https://doi: 10.1158/1078-0432.CCR-15-2762.

Other points of improvement are:

Point 4: Please specify whether specimens from the base of the tongue were included or excluded since this site is naturally rich in lymphatic tissue.

Response 4: In this study we only include cancers located within the oral cavity. Consequently, the base of the tongue was not included as it belongs to the oropharynx.

Point 5: Viral agents that are linked to human tumours, such as human papillomavirus (HPV), has become a great topic of interest during the last years. While HPV probably plays a minor role in cancers arising in the oral cavity it would still be interesting and of additive value to evaluate HPV-status and relate this to B-cell infiltration and other immune profiles.

Response 5: HPV status was evaluated in all cases included in this series by using p16 and p53 immunohistochemistry followed by DNA detection by in situ hybridization (ISH) and polymerase chain reaction (PCR) amplification using the combination of consensus primers MY11/GP6 +. All cases were HPV-negative. These data have previously been published: Rodríguez-Santamarta T, Rodrigo JP, García-Pedrero JM, Álvarez-Teijeiro S, Ángeles Villaronga M, Suárez-Fernández L, Alvarez-Argüelles ME, Astudillo A, de Vicente JC. Prevalence of human papillomavirus in oral squamous cell carcinomas in northern Spain. Eur Arch Otorhinolaryngol. 2016;273(12):4549-4559. doi: 10.1007/s00405-016-4152-9.

It is worth mentioning that the incidence of HPV infection has been extensively studied in different HNSCC patient cohorts in our geographical region (northern Spain), and even for oropharyngeal cancers there is as of yet a low prevalence of HPV incidence (<10%), as we recently reported.

Rodrigo JP, Heideman DA, García-Pedrero JM, Fresno MF, Brakenhoff RH, Díaz Molina JP, Snijders PJ, Hermsen MA. Time trends in the prevalence of HPV in oropharyngeal squamous cell carcinomas in northern Spain (1990-2009). Int J Cancer. 2014 Jan 15;134(2):487-92. doi: 10.1002/ijc.28355.

Rodrigo JP, Hermsen MA, Fresno MF, Brakenhoff RH, García-Velasco F, Snijders PJ, Heideman DA, García-Pedrero JM. Prevalence of human papillomavirus in laryngeal and hypopharyngeal squamous cell carcinomas in northern Spain. Cancer Epidemiol. 2015 Feb;39(1):37-41. doi: 10.1016/j.canep.2014.11.003.

Point 6: Table 3: number of patients is not corresponding with text (page 6, line 136-147).

Response 6: We should clarify that the total number of studied patients slightly varied for the different ratios included in the Table 3. As mentioned in the text (lines 174-177): “It was no possible to calculate the ratios of infiltrating CD20+ B cells and the remaining TILs and macrophage markers for all the cases, since various of these markers showed negative expression (scored as 0). In fact, stromal/tumoral CD20+/ CD8+ ratios could thus be respectively determined in 125 and 118 cases, stromal/tumoral CD20+/ CD4+ ratios respectively determined in 125 and 95 cases, stromal/tumoral CD20+/ FOXP3+ ratios in 106 and 77 cases, stromal/tumoral CD20+/ CD68+ ratios in 125 and 123 cases, and finally, stromal/tumoral CD20+/ CD163+ ratios were determined in 125 and 122 cases, respectively”.

In order to avoid any confusion, the column “number of patients” should better be deleted in Table 3.

Point 7: Table 4: Percentages in text do not correspond with percentages in table.

Response 7: Percentages have been corrected in the text and Table 4.

Point 8: Page 19, line 217: “a stratified survival analyses showed that tumoral CD20+ TILs were significantly associated with prognosis in […]” Please specify high or low density of tumoral CD20+ TILs.

Response 8: We fully agree. This sentence has been modified, and it now reads: “A stratified survival analysis showed that high tumoral CD20+ TILs were significantly associated with a better prognosis in patients”.

Point 9: Page 20, line 254-257: “Noteworthy, we found a significantly reduction in the number of infiltrating CD20+ B cells in T3 and T4 tumors compared with smaller cancer sizes, which suggests that the reduction in the tumoral B lymphocyte infiltration could be associated with tumor progression”.

It could be the other way around; due to the upfront presence of less immune cells and therewith inadequate immunosurveillance of the tumor, the tumor has the opportunity to grow.

Response 9: Many thanks for your valuable comment. Accordingly, this paragraph has been modified (lines 306-308) as follows: “Noteworthy, we found a significant reduction in the number of infiltrating CD20+ B cells in T3 and T4 tumors compared with smaller cancer sizes, which suggests that the reduction in the tumoral B lymphocyte infiltration could be associated with tumor progression. Conversely, it is also plausible that the upfront presence of less immune cells and therewith inadequate immunosurveillance of the tumor, could ultimately favor tumor growth”.

Point 10: Typing/grammatical errors:

  • Page 3, line 10: “[…] as well as tumor PD-L1 and two important CSCs-related factors NANOG and SOX2”.

          PD-L1 expression by tumor cells, or tumoral PD-L1 expression

  • Page 6, line 134: “it was no possible to calculate the ratios of infiltrating CD20+ B cells.
  • Page 6 line 141 and 146: “being this ratio”
  • Page 20, line 294: “Eight-four (67%) patients were active smokers, and 69 (55%) were”

Response 10: Thanks for noticing these errors. All of them have been corrected in this revised version of the manuscript.

Reviewer 3 Report

In the present draft the authors investigated the clinical relevance of infiltrating B lymphocytes in a cohort of 125 OSCC specimens. Data are interesting and are associated with a certain degree of novelty, and the topic discussed and the findings are compatible with the area of interest of the journal.

However, some revision is required. Please find comments below:

  • In the introduction, the authors should better explain what determine/influence the TILs different percentages (lanes 57-59)
  • About the findings in lanes 118-120, can the stroma presence be somehow linked to the presence of driving mutations? In absence of potent driving protumorigenic mutation(s), tumor elements can build a strong dependency from the environment (more stromal elements). Taking this into account, do the authors have the mutation data for their cohort of patients? Also, please comment on that.
  • Lanes 132-133: why the authors observed the correlation only with stromal CD20+ cells and not with the tumoral CD20+ cells? Please comment
  • In table 4, the authors found significative the association between SOX2 and CD20+ intratumoral cells. However, the absolute values are very low. Do the author believe that 1.86 vs 1.33 CD20+ cells per mm2 can have a biological impact? Please comment
  • In figure 2, the authors found a correlation between tumoral CD20+ cells and survival. In the previous data, the relationship was instead with stromal CD20+ cells. How the authors explain this?

Author Response

Reviewer #3

Comments and Suggestions for Authors

In the present draft the authors investigated the clinical relevance of infiltrating B lymphocytes in a cohort of 125 OSCC specimens. Data are interesting and are associated with a certain degree of novelty, and the topic discussed and the findings are compatible with the area of interest of the journal.

Response: We thank the reviewer for the overall positive comments on the interest and novelty of our study.

However, some revision is required. Please find comments below:

Point 1: In the introduction, the authors should better explain what determine/influence the TILs different percentages (lanes 57-59).

Response 1: Further details have now been included in the Introduction (lines 60-68), according to the reviewer’s suggestion.

Tumor-infiltrating lymphocytes in the TME may alter tumor biology, and the type, density, and location of these cells could influence tumor progression. The number of TILs in the TME may be explained by three possibilities. Firstly, the number of immune cells is associated with the immunogenicity of a specific tumor which may induce local activation and proliferation of immune cells. Secondly, it is possible that an increase in B lymphocytes during tumorigenesis and tumor progression may be due to increase in the load of associated antigens on tumor cells, antigens that can induce local activation and proliferation of the immunocytes. Thirdly, the increase in TIL number may reflect enhanced cytokine production by the tumor cells (Hussein, M.R.; Elsers, D.A.H.; Fadel, S.A.; Omar, A.E.M. Immunohistological characterisation of tumour infiltrating lymphocytes in melanocytic skin lesions. J Clin Pathol. 2006, 59, 316–324. doi: 10.1136/jcp.2005.028860).

Point 2: About the findings in lanes 118-120, can the stroma presence be somehow linked to the presence of driving mutations? In absence of potent driving protumorigenic mutation(s), tumor elements can build a strong dependency from the environment (more stromal elements). Taking this into account, do the authors have the mutation data for their cohort of patients? Also, please comment on that.

Response 2: Mutational data were not available for this cohort of patients. The main purpose of our study was to thoroughly investigate the role of B cell infiltration in oral TME in relation to other immune profiles in both stromal and tumoral compartments, jointly evaluating their clinical relevance and impact on patient survival using the same study cohort of 125 OSCC patients. Noteworthy, we have added to this revised version of the manuscript new data from hierarchical clustering analysis further reinforcing the relationships of CD20+ B infiltration with TILs, TAMs and clinical features.

Point 3: Lanes 132-133: why the authors observed the correlation only with stromal CD20+ cells and not with the tumoral CD20+ cells? Please comment.

Response 3: It has recently been reported that the tumor and stroma are distinct in terms of inflammatory mediators associated with tumor progression and disease outcome, and the localization of immune cells has been used as a major descriptor of the TME (Höing, B.; Kanaan, O.; Altenhoff, P.; Petri, R.; Thangavelu, K.;  Schlüter, A.; Lang, S.; Bankfalvi, A.; Brandau, S. Stromal versus tumoral inflammation differentially contribute to metastasis and poor survival in laryngeal squamous cell carcinoma. Oncotarget 2018,9, 8415–8426. https://doi: 10.18632/oncotarget.23865). We herein found that a low B-cell infiltrate in the tumor stroma, but not in tumor nests was associated with a higher tumor stage. These data may suggest that B-cell infiltration in OSCC TME could play an important role in controlling tumor progression.

Point 4: In table 4, the authors found significative the association between SOX2 and CD20+ intratumoral cells. However, the absolute values are very low. Do the author believe that 1.86 vs 1.33 CD20+ cells per mm2 can have a biological impact? Please comment.

Response 4: We unprecedentedly found a significant relationship between SOX2 expression and CD20+ B cell infiltration, and a similar trend was also concordantly observed with NANOG expression. These results suggest a potential link between B cell infiltration and CSC in OSCC; however, its possible biological impact is as of yet unknown.

Point 5: In figure 2, the authors found a correlation between tumoral CD20+ cells and survival. In the previous data, the relationship was instead with stromal CD20+ cells. How the authors explain this?

Response 5: As previously mentioned, tumor and stroma could differently influence tumor progression and disease outcome. In this sense, the ability of B cells to serve as antigen presenting cells may essentially explain the positive outcome of tumoral B cell infiltration that could contribute to intratumoral T cell expansion (Sharonov, G.V.; Serebrovskaya, E.O.; Yuzhakova, D.V.; Britanova, O.V.; Chudakov, D.M. B cells, plasma cells and antibody repertoires in the tumour microenvironment. Nat. Rev. Immunol. 2020,20, 294-307. https://doi: 10.1038/s41577-019-0257-x.

Round 2

Reviewer 1 Report

The authors replied well to the comments from reviewers.

Author Response

Thank you very much

Reviewer 2 Report

The authors have made considerable efforts to improve the manuscript by meeting all points of improvement. Additional analyses are included in the manuscript and some tables have been moved to the supplementals, increasing the ease of reading. Although we were very pleased to see all the efforts that were made, the additive value of the heatmaps is still a bit questionable since no conclusions or thoughts were drawn from these figures besides “a hierarchical clustering analyses is shown”.

Other point of improvement:

Line 280-282: “TLS can be localized around the tumor or even within the tumor itself, and they have one or more germinal centers surrounded by T cells, dendritic and plasma cells, along with lymphatic and blood vessels [53]”

This is incorrect, TLS have different stages of maturation defined by the presence or absence of germinal centers as described in several articles, e.g. “Maturation of tertiary lymphoid structures and recurrence of stage II and III colorectal cancer” by Posch et al.; “Germinal centers determine the prognostic relevance of tertiary lymphoid structures and are impaired by corticosteroids in lung squamous cell carcinoma’ by Silina et al.; “Tumor-infiltrating lymphocytes in the immunotherapy era” by Paijens et al.; “Helper T cell-dominant tertiary lymphoid structures are associated with disease relapse of advanced colorectal cancer” by Yamaguchi et al.

Typing/grammatical errors:

Line 184: “being both ratio’s”

Line 344: “eight-four”

Line 427: “Interestingly, CD20 + TIL density in the tumor nests emerges as an independent prognostic factor in OSCC”

  • Please specify high or low CD20+ TIL density

Author Response

Point 1: The authors have made considerable efforts to improve the manuscript by meeting all points of improvement. Additional analyses are included in the manuscript and some tables have been moved to the supplementals, increasing the ease of reading. Although we were very pleased to see all the efforts that were made, the additive value of the heatmaps is still a bit questionable since no conclusions or thoughts were drawn from these figures besides “a hierarchical clustering analyses is shown”.

Response 1: We thank the reviewer for considering that the changes made and further additional data provided have jointly contributed to improve our work and ease of reading. Results from the heatmaps have further been described (lines 129-132, and also lines 172-176).

Other point of improvement:
Point 2: Line 280-282: “TLS can be localized around the tumor or even within the tumor itself, and they have one or more germinal centers surrounded by T cells, dendritic and plasma cells, along with lymphatic and blood vessels [53]”
This is incorrect, TLS have different stages of maturation defined by the presence or absence of germinal centers as described in several articles, e.g. “Maturation of tertiary lymphoid structures and recurrence of stage II and III colorectal cancer” by Posch et al.; “Germinal centers determine the prognostic relevance of tertiary lymphoid structures and are impaired by corticosteroids in lung squamous cell carcinoma’ by Silina et al.; “Tumor-infiltrating lymphocytes in the immunotherapy era” by Paijens et al.; “Helper T cell-dominant tertiary lymphoid structures are associated with disease relapse of advanced colorectal cancer” by Yamaguchi et al.

Response 2: This has now been corrected according to the reviewer’s comment (lines 289-302), and new references included.

Point 3: Typing/grammatical errors:
Line 184: “being both ratio’s”
Line 344: “eight-four”

Response 3: These errors have now been amended.

Point 4: Line 427: “Interestingly, CD20 + TIL density in the tumor nests emerges as an independent prognostic factor in OSCC”
Please specify high or low CD20+ TIL density

Response 4: This has now been changed both in the Conclusions (lines 443-444), and also similarly at the end of Abstract.